# Five New Pregnane Glycosides from *Gymnema sylvestre* and Their α-Glucosidase and α-Amylase Inhibitory Activities

**DOI:** 10.3390/molecules25112525

**Published:** 2020-05-28

**Authors:** Phan Van Kiem, Duong Thi Hai Yen, Nguyen Van Hung, Nguyen Xuan Nhiem, Bui Huu Tai, Do Thi Trang, Pham Hai Yen, Tran Minh Ngoc, Chau Van Minh, SeonJu Park, Jae Hyuk Lee, Sun Yeou Kim, Seung Hyun Kim

**Affiliations:** 1Institute of Marine Biochemistry, Vietnam Academy of Science and Technology (VAST), 18 Hoang Quoc Viet, Cau Giay, Hanoi 100000, Vietnam; phankiem@yahoo.com (P.V.K.); haiyenk51a@gmail.com (D.T.H.Y.); hungnvd8@yahoo.com (N.V.H.); nxnhiem@yahoo.com (N.X.N.); bhtaiich@gmail.com (B.H.T.); trang2002.imbc@gmail.com (D.T.T.); yeninpc@yahoo.com (P.H.Y.); cvminh@vast.vn (C.V.M.); 2Graduate University of Science and Technology, VAST, 18 Hoang Quoc Viet, Cau Giay, Hanoi 100000, Vietnam; 3Traditional Medicine Administration, Ministry of Health, 138A Giang Vo, Ba Dinh, Hanoi 100000, Vietnam; tmngocvkn@gmail.com; 4Chuncheon Center, Korea Basic Science Institute (KBSI), Chuncheon 24341, Korea; sjp19@kbsi.re.kr; 5College of Pharmacy, Gachon University, 191, Hambakmoero, Yeonsu-gu, Incheon 21936, Korea; wogur6378@naver.com; 6Yonsei Institute of Pharmaceutical Science, College of Pharmacy, Yonsei University, Incheon 21983, Korea; sunnykim@gachon.ac.kr; 7Gachon Institute of Pharmaceutical Science, Gachon University, 191, Hambakmoero, Yeonsu-gu, Incheon 21936, Korea

**Keywords:** *Gymnema sylvestre*, Asclepiadacaea, pregnane, gymsyloside, α-glucosidase, α-amylase

## Abstract

*Gymnema sylvestre*, a medicinal plant, has been used in Indian ayurvedic traditional medicine for the treatment of diabetes. Phytochemical investigation of *Gymnema sylvestre* led to the isolation of five new pregnane glycosides, gymsylosides A–E (**1**–**5**) and four known oleanane saponins, 3β-*O-*β-D-glucopyranosyl (1→6)-β-D-glucopyranosyl oleanolic acid 28-*O-*β-D-glucopyranosyl ester (**6**), gymnemoside-W1 (**7**), 3β-*O-*β-D-xylopyranosyl-(1→6)-β-D- glucopyranosyl-(1→6)-β-D-glucopyranosyl oleanolic acid 28-*O-*β-D-glucopyranosyl ester (**8**), and alternoside XIX (**9**). Their structures were identified based on spectroscopic evidence and comparison with those reported in the literature. All compounds were evaluated for their α-glucosidase and α-amylase inhibitory activities. Compounds **2**–**4** showed significant α-amylase inhibitory activity, with IC_50_ values ranging from 113.0 to 176.2 µM.

## 1. Introduction

*Gymnema sylvestre* (Retz.) R.Br. ex Sm. (Apocynaceae) is a perennial woody climber native to tropical and subtropical regions, such as India, Africa, and southeast Asia. In folk medicine, *G. sylvestre* have been used to treat snake bites, arthritis, digestive, and enhancing laxative [1,2]. Moreover, the plant has been explored for its benefits in blocking sugar craving and reducing sugar consumption. The recent studies have indicated that *G. sylvestre* are potential anti-diabetic plants [3,4]. The bioactive components from this plant include pregnane glycosides [5], triterpene saponins [6,7], and flavonoids [8]. Our previous study reported pregnane glycosides from *Gymnema inodorum* and their α-glucosidase inhibitory activity [9]. As a part of our ongoing investigation on anti-diabetic compounds from Vietnamese plants [10], a methanol extract of the leaves of *G. sylvestre* was found to inhibit α-glucosidase and α-amylase activities. Herein, we report the isolation, structural elucidation of pregnane-type saponins and oleanane saponins and the evaluation of α-glucosidase and α-amylase inhibitory activities of these compounds.

## 2. Results and Discussion

### 2.1. Isolation of Compounds

The methanol extract of the *G. sylvestre* leaves was suspended in water and then partitioned with *n*-hexane, CH_2_Cl_2_ and EtOAc to obtain four layers. The CH_2_Cl_2_ and water extracts were chromatographed using combined silica gel and RP-18 columns. The fractions were further purified by HPLC to give five new pregnane glycosides and four known compounds (Figure 1 and Appendix A).

### 2.2. Compound Identification

Compound **1** was obtained as a white amorphous powder. Its molecular formula was determined as C_47_H_76_O_17_ by HRESIMS ion at *m*/*z* 935.4986 [M + Na]^+^ (calcd for [C_47_H_76_O_17_Na]^+^, 935.4975). The ^1^H-NMR spectrum showed proton signals of three methyl groups at *δ*_H_ 1.13 (3H, s), 1.53 (3H, s), and 1.05 (3H, d, *J* = 6.4 Hz), one olefinic proton at *δ*_H_ 5.32 (1H, br s), which represented a pregnane aglycone. Two methyl groups at *δ*_H_ 1.80 (3H, d, *J* = 7.2 Hz) and 1.85 (3H, s) and one olefinic proton at *δ*_H_ 6.98 (1H, q, *J* = 7.2 Hz), suggested the presence of a tigloyl moiety. Three anomeric protons [*δ*_H_ 4.84 (br d, *J* = 9.6 Hz), 4.57 (br d, *J* = 9.2 Hz), and 4.41 (br d, *J* = 8.0 Hz), three methoxy groups [*δ*_H_ 3.39, 3.42, and 3.60 (each 3H, s)], together with three secondary methyl groups [*δ*_H_ 1.19 (3H, d, *J* = 6.0 Hz), 1.25 (d, *J* = 6.0 Hz), and 1.35 (d, *J* = 6.0 Hz)], confirmed the presence of three sugar units. The ^13^C NMR and DEPT spectra indicated that **1** contained one carbonyl, seven non-protonated carbons (three oxygenated), nineteen methines (sixteen oxygenateds), nine methylenes, and eleven methyl carbons (three methoxys). The ^1^H and ^13^C-NMR spectroscopic data suggested that **1** was a pregnane glycoside [11]. Of these, 21 carbons were assigned to the pregnane skeleton, 5 to one tigloyl moiety, and 21 to a trisaccharide moiety. The HMBC correlations between H-19 (*δ*_H_ 1.13) and C-1 (*δ*_C_ 39.8)/C-5 (*δ*_C_ 140.0)/C-9 (*δ*_C_ 44.7)/C-10 (*δ*_C_ 38.0) suggested the position of a double bond at C-5/C-6. The HMBC correlations between H-18 (*δ*_H_ 1.53) and C-13 (*δ*_C_ 57.6)/C-14 (*δ*_C_ 89.3)/C-17 (*δ*_C_ 89.1); between H-21 (*δ*_H_ 1.05) and C-17 (*δ*_C_ 89.1)/C-20 (*δ*_C_ 71.5); and between H-6 (*δ*_H_ 5.32)/H-7 (*δ*_H_ 2.11)/H-9 (*δ*_H_ 1.49) and C-8 (*δ*_C_ 74.9) demonstrated the positions of hydroxyl groups at C-8, C-14, C-17, and C-20 (Figure 2). The constitution of the aglycone of **1** was demonstrated by the analysis of NOESY observations and similar reported-structures [11].

The aglycone of **1** was supposed to have the same configurations as those of gymnepregoside F and 12-*O*-(*E*)-cinnamoylgymnepregoside F from *G. sylvestre* [12], biogenetic derivatives of **1** at C-3, C-8, C-12, C-14, C-17, and C-20. In addition, the alkaline hydrolysis of **1** gave sarcostin, ((20*S*)-3β,8β,12β,14β,17β,20-hexahydroxypregn-5-ene) [13]. The multiplicity of H-12 [*δ*_H_ 4.68 (dd, *J* = 4.0, 11.6 Hz)] suggested that the configuration of H-12 was *axial* (α-configuration, Figure 2). The NOESY correlations between H-3 (*δ*_H_ 3.50) and H_α_-1 (*δ*_H_ 1.09)/ H_α_-4 (*δ*_H_ 2.33), and between H-12 (*δ*_H_ 4.68) and H-9 (*δ*_H_ 1.49)/H_α_-15 (*δ*_H_ 1.88) suggested the configurations of the oxygenated groups at C-3 and C-12, the hydroxy groups at C-8 and C-14 to be β. The HMBC correlations between Tig H-5 (*δ*_H_ 1.85) and Tig C-1 (*δ*_C_ 169.1)/Tig C-2 (*δ*_C_ 130.0)/Tig C-3 (*δ*_C_ 139.5) and between Tig H-4 (*δ*_H_ 1.80) and Tig C-2/ Tig C-3 and NOESY correlations between Tig H-4 (*δ*_H_ 1.80) and Tig H-5 (*δ*_H_ 1.85) suggested the presence of (*E*)-tigloyl moiety. In addition, the position of this moiety at C-12 was confirmed by HMBC correlation from H-12 (*δ*_H_ 4.68) to Tig C-1 (*δ*_C_ 169.1). Acid hydrolysis of **1** gave three monosaccharides, which were identified as D-cymarose [14], D-oleandrose [14], and D-thevetose [15], by comparing their specific rotation with those reported [16]. The large coupling constants between H-1 and H-2 of monosaccharide moieties and also HMBC correlations between Thv H-1 (*δ*_H_ 4.41) and Ole C-4 (*δ*_C_ 84.1), Ole H-1 (*δ*_H_ 4.57) and Cym C-4 (*δ*_C_ 83.8), and between Cym H-1 (*δ*_H_ 4.84) and aglycone C-3 (*δ*_H_ 79.2) indicated the sugar linkages as β-D-thevetopyranosyl-(1→4)-β-D-oleandropyranosyl- (1→4)-β-D-cymaropyranoside and at C-3 of aglycone. Based on the above evidence, the structure of **1** was elucidated as (20*S*)-12β-tigloyloxy-3β,8β,14β,17β,20-pentahydroxypregn-5-ene 3-*O*-β-D-thevetopyranosyl- (1→4)-β-D-oleandropyranosyl-(1→4)-β-D-cymaropyranoside, a new compound named gymsyloside A.

The ^1^H and ^13^C-NMR spectra of **2** exhibited a pregnane aglycone, one tigloyl unit, and three sugar units (Table 1). In addition, the NMR data of **2** were similar to those of gymsyloside A (**1**), except for the difference of sugar unit at Ole C-4: D-thevetose replaced by 6-deoxy-3-*O*-methyl-D-allose. Acid hydrolysis of **2** confirmed the presence of D-cymarose, D-oleandrose, and 6-deoxy-3-*O*-methyl-D-allose as sugar components. Furthermore, the ^1^H and ^13^C-NMR data of **2** showed the sugar units as β-D-cymaropyranosyl, β-D-oleandropyranosyl, and 6-deoxy-3-*O*-methyl-β-D-allopyranose. The HMBC correlations between All H-1 (*δ*_H_ 4.70) and Ole C-4 (*δ*_C_ 84.0), Ole H-1 (*δ*_H_ 4.56) and Cym C-4 (*δ*_C_ 83.8), and between Cym H-1 (*δ*_H_ 4.85) and aglycone C-3 (*δ*_C_ 79.3) confirmed the sugar linkages to be 3-*O*-6-deoxy-3-*O*-methyl-β-D-allopyranosyl-(1→4)-β-D-oleandropyranosyl-(1→4)-β-D-cymaropyranoside. Consequently, compound **2** was elucidated to be (20*S)*-12β-tigloyloxy-3β,8β,14β,17β,20-pentahydroxypregn-5-ene 3-*O*-6-deoxy-3-*O*-methyl-β-D-allopyranosyl-(1→4)-β-D-oleandropyranosyl-(1→4)-β-D-cymaropyranoside, a new compound named gymsyloside B.

The HRESIMS of **3** gave a pseudo-molecular ion peak at *m*/*z* 1079.5769 [M + Na]^+^, corresponding to the molecular formula of C_54_H_88_O_20_. The ^1^H and ^13^C-NMR spectra of **3** showed the presence of one pregnane aglycone, four sugar units, and one tigloyl unit (Table 1). The NMR data of **3** were compared to gymsyloside A (**1**) and found the addition of one sugar unit in the sugar linkages. Acid hydrolysis of **3** gave three monosaccharides, which were identified as D-cymarose, D-oleandrose, and D-thevetose. The tetrasaccharide was determined to be β-D-thevetopyranosyl-(1→4)-β-D-oleandropyranosyl-(1→4)-β-D-cymaropyranosyl-(1→4)-β-D-cymaropyranoside, by the analysis of HMBC and COSY correlations. The location of sugar linkages at C-3 was confirmed by the HMBC correlation between Cym I H-1 (*δ*_H_ 4.84) and C-3 (*δ*_C_ 79.3). Consequently, the structure of **3** was determined to be (20*S*)-12β-tigloyloxy-3β,8β,14β,17β, 20-pentahydroxypregn-5-ene 3-*O*-β-D-thevetopyranosyl-(1→4)-β-D-oleandropyranosyl-(1→4)-β-D-cymaropyranosyl-(1→4)-β-D-cymaropyranoside, a new compound named gymsyloside C.

The molecular formula of **4** was determined as C_54_H_88_O_20_ by the HRESIMS. The ^1^H and ^13^C-NMR data (Table 2) indicated that the structure of **4** was similar to those of **3**, except for the difference of monosaccharide at Ole C-4. The sugar components were found to be similar to those of **2** (D-cymarose, D-oleandrose, and 6-deoxy-3-*O*-methyl-D-allose) [17]. Moreover, the sugar linkages, 3-*O*-6-deoxy-3-*O*-methyl-β-D-allopyranosyl-(1→4)-β-D-oleandropyranosyl-(1→4)-β-D-cymaropyranosyl-(1→4)-β-D-cymaropyranoside was confirmed by the HMBC correlations from All H-1 (*δ*_H_ 4.70) to Ole C-4 (*δ*_C_ 83.7), Ole H-1 (*δ*_H_ 4.56) to Cym II C-4 (*δ*_C_ 83.8), and from Cym II H-1 (*δ*_H_ 4.77) to Cym I C-4 (*δ*_C_ 83.8). Similar to those of **1**–**3**, the aglycone was found as (20*S)*-12β-tigloyloxy-3β,8β,14β,17β, 20-pentahydroxypregn-5-ene. Consequently, the structure of **4** was determined as (20*S)*- 12β-tigloyloxy-3β,8β,14β,17β,20-pentahydroxypregn-5-ene 3-*O*-6-deoxy-3-*O*-methyl-β-D- allopyranosyl-(1→4)-β-D-oleandropyranosyl-(1→4)-β-D-cymaropyranosyl-(1→4)-β-D-cymaropyranoside and named gymsyloside D.

The molecular formula of **5,** C_53_H_86_O_22_ was determined by the HRESIMS pseudo-ion at *m*/*z* 1097.5530 [M + Na]^+^. The ^1^H and ^13^C NMR data of **5** were similar to **2**, except for an additional sugar unit at Thv C-4 (Table 2). The sugar moieties were determined as D-cymarose [14], D-oleandrose [14], D-thevetose [15], and D-glucose [17] by acid hydrolysis. The HMBC correlations between Glc H-1 (*δ*_H_ 4.41) and Thv C-4 (*δ*_C_ 82.9), Thv H-1 (*δ*_H_ 4.43) and Ole C-4 (*δ*_C_ 84.1), Ole H-1 (*δ*_H_ 4.57) and Cym C-4 (*δ*_C_ 83.8), and between Cym H-1 (*δ*_H_ 4.84) and aglycone C-3 (*δ*_C_ 79.3) confirmed the sequence of sugar linkages, previously reproted from *G. sylvestre* [5]. Thus, compound **5** was characterized as (20*S)*-12β-tigloyloxy-3β,8β,14β,17β,20-pentahydroxypregn-5-ene 3-*O*-β-D-glucopyranosyl-(1→4)-β-D-thevetopyranosyl-(1→4)-β-D-oleandropyranosyl-(1→4)-β-D-cymaropyranoside and named gymsyloside E.

The known compounds were identified as 3β-*O-*β-D-glucopyranosyl (1→6)-β-D-glucopyranosyl oleanolic acid 28-*O-*β-D-glucopyranosyl ester (**6**) [7], gymnemoside-W1 (**7**) [18], 3β-*O-*β-D-xylopyranosyl-(1→6)-β-D-glucopyranosyl-(1→6)-β-D-glucopyranosyl oleanolic acid 28-*O-*β-D-glucopyranosyl ester (**8**) [7], and alternoside XIX (**9**) [18]. These compounds were already reported from *G. sylvestre*. Thus, for oleanane saponins, the main components could be hightly species-specific of *G. sylvestre* [7,18]. In addition, new pregnane glycosides were also found in *G. alternifolium* [19], *G. sylvestre* [5], and *G. griffithii* [15]. Five new pregnane glycosides in this report will contribute specific compounds in *Gymnema* genus.

### 2.3. α-Glucosidase and α-Amylase Inhibitory Activities

All compounds were evaluated for the α-glucosidase and α-amylase inhibitory assays. Compound **4** showed the weak α-glucosidase inhibitory activity (16.4 ± 2.3%) at the concentration of 200 µM, compared tos the positive control, acarbose (inhibition percentage of 57.8 ± 3.2% at the concentration of 155 µM) (Figure 3). Compounds **2**–**4** showed α-amylase inhibitory activity with inhibition percent ranging from 57.9% to 66.8% at the concentration of 200 µM (Figure 4). In the subsequent concentration-dependent assay, compounds **2**, **3**, and **4** showed significant α-amylase inhibitory activity, with IC_50_ values of 175.8 ± 2.3, 162.2 ± 2.7, and 113.0 ± 0.7 µM, respectively, compared to positive control, acarbose (IC_50_ value of 72.4 ±0.8 µM). This is the first report of α-glucosidase and α-amylase inhibitory activities of compounds from *G. sylvestre.* Recent reports have shown insulin secretion stimulation of *G. sylvestre* extract [4], antihyperglycemic effects of gmynemic acids [20], α-glucosidase and α-amylase inhibitory activities of pregnane glycosides from *G. latifolium* [11]. Previous studies have indicated that pregnane glycosides from *G. griffithii* showed moderate α-glucosidase inhibitory activity [15]. Russelioside B, a pregnane glycoside, possessed antidiabetic and antihyperlipidemic effect in streptozotocin induced diabetic rats [21]. Therefore, the results suggest that the discovery of pregane glycosides may increase the possibility of finding antidiabetic agents.

## 3. Materials and Methods

### 3.1. General

All NMR spectra were recorded on an Agilent 400-MR-NMR (Agilent technologies, Santa Clara, CA, USA) spectrometer operated at 400 and 100 MHz for hydrogen and carbon, respectively. Data processing was carried out with the MestReNova ver.6.0.2 program. HRESIMS spectra were obtained using an AGILENT 6550 iFunnel Q-TOF LC/MS system (Agilent technologies, Santa Clara, CA, USA). Optical rotations were determined on a Jasco DIP-370 automatic polarimeter. Preparative HPLC was carried out using an AGILENT 1200 HPLC system. Column chromatography was performed on silica-gel (Kieselgel 60, 70–230 mesh and 230–400 mesh, Merck) or YMC RP-18 resins (30–50 µm, Fuji Silysia Chemical Ltd., Aichi, Japan). For thin layer chromatography (TLC), a pre-coated silica-gel 60 F254 (0.25 mm, Merck, Darmstadt, Germany) and RP-18 F254S plates (0.25 mm, Merck, Darmstadt, Germany) were used.

### 3.2. Plant Material

The leaves of *Gymnema sylvestre* (Retz.) R.Br. ex Sm. were collected in Hai Loc, Hai Hau, Nam Dinh in November, 2015, and identified by Dr. Nguyen The Cuong, Institute of Ecology and Biological Resources. A voucher specimen (NCCT-P20) was deposited at the Herbarium Institute of Marine Biochemistry, VAST.

### 3.3. Extraction and Isolation

The dried powders of *G. sylvestre* leaves (4.0 kg) were sonicated with hot methanol (3 times × 10 L, each 3 h) to give MeOH extract (450 g), after evaporation of the solvent. The MeOH extract was suspended in water and successively partitioned with *n*-hexane, CH_2_Cl_2_ and EtOAc to obtain the *n*-hexane (GS1, 47.0 g), CH_2_Cl_2_ (GS2, 60.0 g), EtOAc extracts (GS3, 27.0 g) and H_2_O layer (GS4). GS2 was chromatographed on a silica gel column (180.0 g, silica gel) eluting with gradient solvent of *n*-hexane:acetone (40:1, 20:1, 10:1, 5:1, 1:1, and 0:1, *v*/*v*, each 2 L), to give seven fractions, GS2A-GS2G. The GS2F fraction was chromatographed on a silica gel column eluting with CHCl_3_:MeOH (11:1, *v*/*v*) to give four fractions, GS2F1-GS2F4. GS2F1 was chromatographed on a RP-18 column using MeOH:H_2_O (4:1, *v*/*v*) as a solvent, to give five fractions, GS2F1A-GS2F1E. Compounds **1** (17.0 mg, t_R_ 38.5 min) and **2** (9.0 mg, t_R_ 42.1 min) were yielded from GS2F1B fraction using HPLC system: J’sphere H-80 column (150 × 20 mm), flow rate of 3 mL/min, and solvent condition of 40% acetonitrile in water. Compounds **3** (5.0 mg, t_R_ 44.7 min) and **4** (5.0 mg, t_R_ 49.4 min) were yielded from GS2F1D fraction on J’sphere H-80 column (150 × 20 mm), flow rate of 3 mL/min, and solvent condition of 40% acetonitrile in water. GS2G was chromatographed on RP-18 column eluting with MeOH:H_2_O (4:1, *v*/*v*) to give four smaller fractions, GS2G1-GS2G4. Finally, GS2G3 was chromatographed on J’sphere H-80 column (150 × 20 mm), flow rate of 3 mL/min, and solvent condition of 35% acetonitrile in water to yield compound **5** (14.0 mg, t_R_ 39.7 min). GS4 was chromatographed on a Diaion column and eluted with H_2_O then increased concentrations of MeOH in H_2_O, to obtain sub-fractions, GS4A-GS4C. GS4C was chromatographed on a silica gel column eluting with a gradient of CHCl_3_:MeOH (20:1, 10:1, 5:1, 1:1, *v*/*v*) to give smaller fractions, GS4C1-GS4C4. GS4C4 was chromatographed on an RP-18 CC eluting with MeOH:water (2:1, *v*/*v*) to give smaller fractions, GS4C4A-GS4C4E. GS4C4B was chromatographed on an RP-18 column eluting with acetone:H_2_O (0.8:1, *v*/*v*) to yield **7** (5.0 g) and **9** (40.0 mg). GS4C4E was chromatographed on an RP-18 column eluting with acetone:H_2_O (1:1, *v*/*v*), to yield **6** (100.0 mg) and **8** (5.0 mg).

#### 3.3.1. Gymsyloside A (**1**)

White amorphous powder; ${\left[\alpha \right]}_{D}^{25}$ −20.0 (*c* 0.1, MeOH); C_47_H_76_O_17_, HRESIMS *m*/*z*: 935.4986 [M + Na]^+^ (calcd for [C_47_H_76_O_17_Na]^+^, 935.4975); ^1^H (CD_3_OD, 400 MHz) and ^13^C-NMR (CD_3_OD, 100 MHz) data, see Table 1.

#### 3.3.2. Gymsyloside B (**2**)

White amorphous powder; ${\left[\alpha \right]}_{D}^{25}$ + 35.0 (*c* 0.1, MeOH); C_47_H_76_O_17_, HRESIMS *m*/*z*: 935.4996 [M + Na]^+^ (calcd for [C_47_H_76_O_17_Na]^+^, 935.4975); ^1^H (CD_3_OD, 400 MHz) and ^13^C-NMR (CD_3_OD, 100 MHz) data, see Table 1.

#### 3.3.3. Gymsyloside C (**3**)

White amorphous powder; ${\left[\alpha \right]}_{D}^{25}$ + 80.0 (*c* 0.1, MeOH); C_54_H_88_O_20_, HRESIMS *m*/*z*: 1079.5769 [M + Na]^+^ (calcd for [C_54_H_88_O_20_Na]^+^, 1079.5769); ^1^H (CD_3_OD, 400 MHz) and ^13^C-NMR (CD_3_OD, 100 MHz) data, see Table 1.

#### 3.3.4. Gymsyloside D (**4**)

White amorphous powder; ${\left[\alpha \right]}_{D}^{25}$+ 58.7 (*c* 0.1, MeOH); C_54_H_88_O_20_, HRESIMS *m*/*z*: 1079.5778 [M + Na]^+^ (calcd for [C_54_H_88_O_20_Na]^+^, 1079.5769); ^1^H (CD_3_OD, 400 MHz) and ^13^C-NMR (CD_3_OD, 100 MHz) data, see Table 2.

#### 3.3.5. Gymsyloside E (**5**)

White, amorphous powder; ${\left[\alpha \right]}_{D}^{25}$+ 54.0 (*c* 0.1, MeOH); C_53_H_86_O_22_, HRESIMS *m*/*z*: 1097.5530 [M + Na]^+^ (calcd for [C_53_H_86_O_22_Na]^+^, 1097.5503); ^1^H (CD_3_OD, 400 MHz) and ^13^C NMR (CD_3_OD, 100 MHz) data, see Table 2.

### 3.4. Acid Hydrolysis

Each compound (**1**–**5**, 3.0 mg) was separately dissolved in 1.0 N HCl (dioxane—H_2_O, 1:1, *v*/*v*, 1.0 mL) and heated to 80 °C in a water bath for 3 h. The acidic solution was dried under N_2_ overnight. After extraction with CHCl_3_, the aqueous layer was dried using N_2_ to give aqueous residue (A). The aqueous residue (A) was separated by silica gel CC eluting with CH_2_Cl_2_–MeOH (10:1, *v*/*v*) and then further fractionated by RP-18 CC using a solvent gradient of MeOH–H_2_O (6:4, 7:3, and 8:2, *v*/*v*), to give the monosaccharides (50% yield). The specific rotations of these sugars were determined. The specific rotations (${\left[\alpha \right]}_{D}^{25}$) of sugars was determined after dissolving in H_2_O for 24 h and compared to the literature (lit):D-cymarose: found +50.1 (*c* 0.4, H_2_O), lit +51.8 [14]; D-oleandrose: found −12.1 (*c* 0.4, H_2_O), lit +11.7 [14]; D-thevetose: found +40.3 (*c* 0.4, H_2_O); lit +42.3 [15]; 6-deoxy-3-*O*-methyl-D-allose: found +10.9 (*c* 0.4, H_2_O); lit +10.0 [17]; D-glucose: found + 49.2 (*c* 0.4, H_2_O); lit +48.0 [17]. Based on the above evident, sugar components were found in: compounds **1** and **3**: D-cymarose, D-oleandrose, and D-thevetose; compounds **2** and **4**: D-cymarose, D-oleandrose, and 6-deoxy-3-*O*-methyl-D-allose; compound **5**: D-cymarose, D-oleandrose, D-thevetose, and D-glucose.

### 3.5. Alkaline Hydrolysis

A solution of compound **1** (8 mg) in 1.0 mL of 5% KOH/MeOH was heated at 40 °C four 4 h and then neutralized with HCl 0.1 M. After that, the solution was partitioned with CHCl_3_ to give CHCl_3_ layer. CHCl_3_ layer was separated on HPLC system: J’sphere H-80 column (150 × 20 mm), solvent condition of 55% acetonitrile, to give sarcostin (54% yield). In a similar way, sarcostin was found as aglycone of compounds **2**–**5.**

### 3.6. α-Glucosidase Inhibitory Assay

The α-glucosidase (G0660-750UN, Sigma-Aldrich, St. Louis, MO) enzyme inhibition assay was performed according to the previously described method [11]. The sample solution (2 mL dissolved in dimethyl sulfoxide (DMSO)) and 0.5 U/mL α-glucosidase (40 mL) were mixed in 120 mL of 0.1 M phosphate buffer (pH 7.0). After 5 min pre-incubation, 5 mM p-nitrophenyl-α-D-glucopyranoside solution (40 mL) was added, and the solution was incubated at 37 °C for 30 min. The absorbance of released 4-nitrophenol was measured at 405 nm by using a microplate reader (Molecular Devices, Sunnyvale, CA, USA).

### 3.7. α-Amylase Inhibitory Assay

The α-amylase (A8220, Sigma-Aldrich, St. Louis, MO, USA) enzyme inhibitory activity was measured using the reported method [11]. Substrate was prepared by boiling 100 mg potato starch in 5 mL phosphate buffer (pH 7.0) for 5 min, then cooling to room temperature. The samples (2 mL dissolved in DMSO) and substrate (50 mL) were mixed in 30 mL of 0.1 M phosphate buffer (pH 7.0). After 5 min pre-incubation, 5 mg/mL α-amylase solution (20 mL) was added, and the solution was incubated at 37 °C for 15 min. The reaction was stopped by adding 50 mL 1 M HCl and then 50 mL iodine solution was added. The absorbances were measured at 650 nm by a microplate reader.

## 4. Conclusions

In summary, five new pregnane glycosides and four known oleanane saponins were isolated and identified from *G. sylvestre*. Besides oleanane saponins-hightly species-specific of *G. sylvestre,* new pregnane glycosides in this report will provide secondary metabolisms as specific compounds in *Gymnema* genus. Compound **4** showed the weak α-glucosidase inhibitory activity. Compounds **2**, **3**, and **4** showed significant α-amylase inhibitory activity. This is the first report of α-amylase and α-glycosidase inhibitory activities of compounds from *G. sylvestre.*

## Figures and Tables

**Figure 1 molecules-25-02525-f001:**
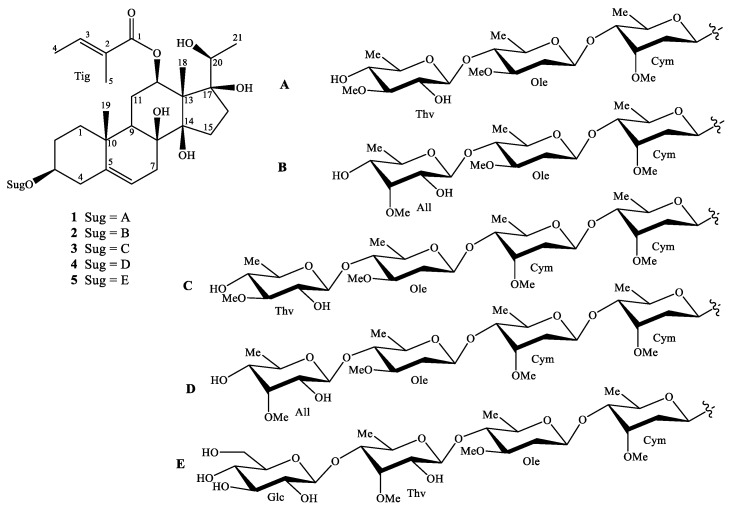
Chemical structures of compounds of **1**–**5**.

**Figure 2 molecules-25-02525-f002:**
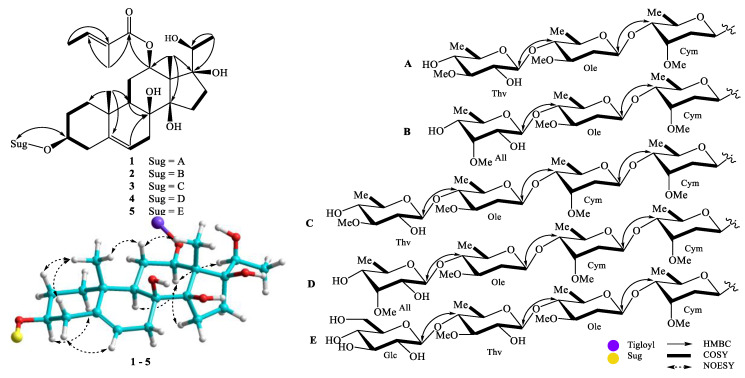
The key HMBC, COSY, and NOESY correlations of compounds **1**–**5.**

**Figure 3 molecules-25-02525-f003:**
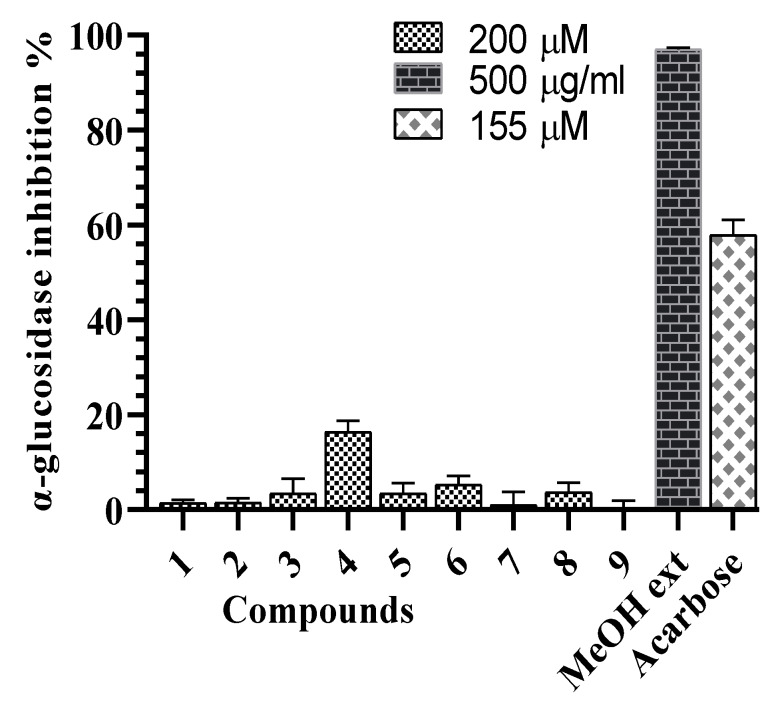
α-Glucosidase inhibitory effects of the *G. sylvestre* extract and compounds **1**–**9**.

**Figure 4 molecules-25-02525-f004:**
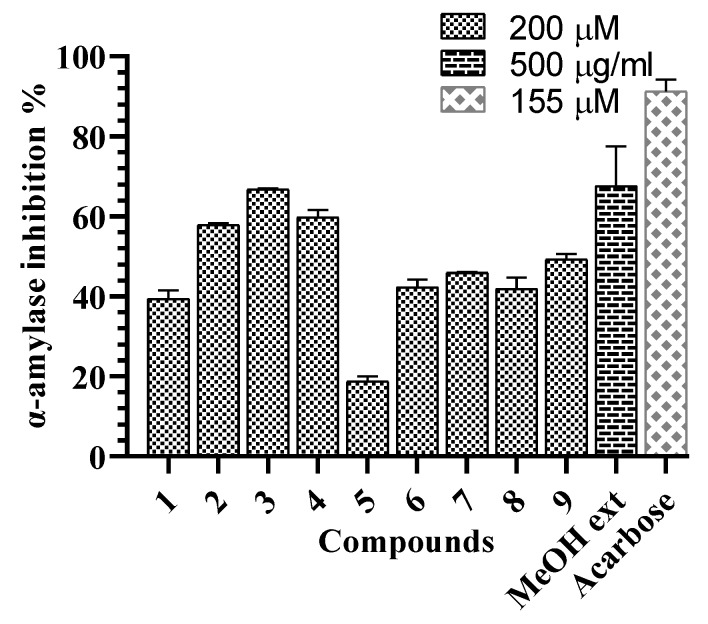
α-Amylase inhibitory effects of the *G. sylvestre* extract and compounds **1**–**9**.

**Table 1 molecules-25-02525-t001:** NMR spectroscopic data for compounds **1**–**3** in CD_3_OD.

C		1		2		3
	*δ* _C_	*δ*_H_ (mult., *J*, in Hz)	*δ* _C_	*δ*_H_ (mult., *J*, in Hz)	*δ* _C_	*δ*_H_ (mult., *J*, in Hz)
1	39.8	1.09, m, α/1.80, m, *β*	39.8	1.09, m, α/1.79, m, *β*	39.8	1.09, m, α/1.80, m
2	30.2	1.57, m, *β*/1.84, m, α	30.2	1.56, m, *β*/1.84, m, α	30.1	1.57, m, *β*/1.83, m, α
3	79.2	3.50, m	79.3	3.50, m	79.3	3.49, m
4	39.8	2.20, m, *β*/2.33, m, α	39.8	2.20, m, *β*/2.33, m, α	39.8	2.19, m, *β*/2.33, m, α
5	140.0	-	140.0	-	140.0	-
6	120.0	5.32, br s	120.0	5.32, br s	119.9	5.33, br s
7	35.2	2.11, m	35.2	2.12, m	35.2	2.12, m
8	74.9	-	75.0	-	75.0	-
9	44.7	1.49, m	44.7	1.48, m	44.7	1.49, m
10	38.0	-	38.0	-	38.0	-
11	26.0	1.63, m, α/2.00, m, *β*	26.0	1.62, m, α/2.00, m, *β*	25.9	1.62, m, α/2.00, m, *β*
12	75.1	4.68, dd, 4.0, 11.6)	75.2	4.68, m	75.2	4.70, dd, 4.0, 9.6)
13	57.6	-	57.6	-	57.6	-
14	89.3	-	89.3	-	89.2	-
15	34.3	1.83, m, *β*/1.88, m, α	34.3	1.82, m, *β*/1.91, m, α	34.3	1.81, m, *β*/1.89, m, α
16	33.5	1.74, m	33.5	1.75, m	33.5	1.74, m
17	89.1	-	89.2	-	89.3	-
18	11.2	1.53, s	11.2	1.53, s	11.2	1.52, s
19	18.5	1.13, s	18.5	1.14, s	18.5	1.14, s
20	71.5	3.45, m	71.5	3.44, m	71.6	3.46, m
21	18.9	1.05, d (6.4)	18.9	1.04, d (6.4)	18.9	1.06, d (6.4)
	**Tig**		**Tig**		**Tig**	
1	169.1	-	169.2	-	169.1	-
2	130.0	-	130.1	-	130.1	-
3	139.5	6.98, q (7.2)	139.6	6.98, q (7.2)	139.6	7.00, q (6.8)
4	14.5	1.80, d (7.2)	14.5	1.80, d (7.2)	14.5	1.81, d (6.8)
5	12.2	1.85, s	12.1	1.85, s	12.1	1.87, s
	**Cym**		**Cym**		**Cym I**	
1	97.2	4.84, br d (9.6)	97.2	4.85, br d (9.6)	97.2	4.84, br d, 9.6)
2	36.6	1.52, m, *a*/2.05, m, *e*	36.7	1.53, m, *a*/2.04, m, *e*	36.6	1.55, m, *a*/2.05, m, *e*
3	78.5	3.82, m	78.5	3.83, m	78.5	3.82, m
4	83.8	3.24, m	83.8	3.24, m	83.8	3.23, m
5	69.9	3.79, m	70.0	3.79, m	70.1	3.79, m
6	18.5	1.19, d (6.0)	18.5	1.19, d (6.0)	18.5	1.18, d (6.4)
3-OMe	58.5	3.42, s	58.5	3.42, s	58.5	3.42, s
	**Ole**		**Ole**		**Cym II**	
1	102.6	4.57, br d (9.2)	102.6	4.56, br d (6.8)	101.2	4.78, br d (9.6)
2	37.6	1.40, m, *a*/2.30, m, *e*	37.5	1.39, m, *a*/2.30, m, *e*	36.4	1.55, m, *a*/2.10, m, *e*
3	80.2	3.36, m	80.4	3.36, m	78.6	3.82, m
4	84.1	3.18, m	84.0	3.17, m	83.9	3.23, m
5	72.5	3.36, m	72.6	3.35, m	69.9	3.79, m
6	18.9	1.35, d (6.0)	19.0	1.34, d (6.0)	18.6	1.21, d (6.0)
3-OMe	57.6	3.39, s	57.5	3.39, s	58.4	3.42, s
	**Thv**		**All**		**Ole**	
1	104.3	4.41, br d (8.0)	102.2	4.70, br d (7.6)	102.6	4.58, br d (8.4)
2	75.6	3.18, m	73.6	3.29, m	37.6	1.40, m, *a*)/2.30, m, *e*)
3	87.7	3.00, t (6.8)	83.8	3.60, m	80.2	3.36, m
4	76.6	3.00, t (6.8)	75.0	3.16, m	84.1	3.18, m
5	73.2	3.25, m	71.2	3.64, m	72.5	3.35, m
6	18.1	1.25, d (6.0)	18.5	1.21, d (6.0)	18.9	1.36, d (5.6)
3-OMe	61.0	3.60, s	62.5	3.58, s	57.5	3.41, s
					**Thv**	
1	-	-	-	-	104.3	4.42, d (8.0)
2	-	-	-	-	75.6	3.18, m
3	-	-	-	-	87.7	3.00, m
4	-	-	-	-	76.6	3.00, m
5	-	-	-	-	73.2	3.25, m
6	-	-	-	-	18.1	1.27, d (6.0)
3-OMe	-	-	-	-	61.1	3.60, s

Assignments were done by Heteronuclear Single Quantum Coherence (HSQC), Heteronuclear Multiple Bond Correlation (HMBC), Correlation Spectroscopy (COSY), and Rotating frame Overhauser Effect Spectroscopy (ROESY) experiments. Tig, Tigloyl; Cym, β-d-cymaropyranosyl; Ole, β-d-oleandropyranosyl; Thv, β-d-thevetopyranosyl; All, 6-deoxy-3-*O*-methyl-β-d-allopyranosyl; *a*, axial; *e*, equatorial; *α*, atoms or groups laying below the plane of structure; *β*, atoms or groups laying above the plane of structure.

**Table 2 molecules-25-02525-t002:** NMR spectroscopic data for compounds **4** and **5** in CD_3_OD.

C		4		5
	*δ* _C_	*δ*_H_ (mult., *J*, in Hz)	*δ* _C_	*δ*_H_ (mult., *J*, in Hz)
1	39.8	1.07, m, α/1.79, m, *β*	39.8	1.08, m, α/1.80, m, *β*
2	30.2	1.57, m, *β*/1.83, m, α	30.2	1.58, m, *β*/1.84, m, α
3	79.3	3.50, m	79.3	3.50, m
4	39.8	2.18, m, *β*/2.33, m, α	39.8	2.20, m, *β*/2.33, m, α
5	140.0	-	140.0	-
6	120.0	5.32, br s	120.0	5.32, br s
7	35.2	2.11, m	35.2	2.11, m
8	74.9	-	74.9	-
9	44.7	1.49, m	44.7	1.49, m
10	38.0	-	38.0	-
11	26.0	1.63, m, α/2.00, m, *β*	25.9	1.63, m, α/2.00, m, *β*
12	75.1	4.67, m	75.1	4.68, m
13	57.6	-	57.6	-
14	89.3	-	89.3	-
15	34.3	1.80, m, *β*/1.88, m, α	34.3	1.83, m, *β*/1.90, m, α
16	33.5	1.74, m	33.5	1.75, m
17	89.1	-	89.1	-
18	11.2	1.53, s	11.2	1.53, s
19	18.6	1.13, s	18.6	1.12, s
20	71.5	3.46, m	71.5	3.45, m
21	19.0	1.05, d (6.0)	18.9	1.05, d (6.4)
	**Tig**		**Tig**	
1	169.1	-	169.1	-
2	130.1	-	130.1	-
3	139.5	6.98, q (7.2)	139.5	6.98, q (7.2)
4	14.6	1.81, d (7.2)	14.5	1.80, d (7.2)
5	12.2	1.85, s	12.2	1.85, s
	**Cym I**		**Cym**	
1	97.2	4.83, br d (9.6)	97.2	4.84, br d (9.6)
2	36.6	1.54, m, *a*/2.04, m, *e*	36.7	1.53, m, *a*/2.05, m, *e*
3	78.5	3.82, m	78.5	3.81, m
4	83.8	3.24, m	83.8	3.24, m
5	69.8	3.78, m	69.9	3.80, m
6	18.3	1.17, d (6.4)	18.4	1.19, d (6.4)
3-OMe	58.4	3.41, s	58.5	3.42, s
	**Cym II**		**Ole**	
1	101.2	4.77, br d (9.6)	102.6	4.57, br d (9.2)
2	36.4	1.54, m, *a*/2.04, m, *e*	37.6	1.40, m, *a*/2.30, m, *e*
3	78.4	3.82, m	80.3	3.35, m
4	83.8	3.24, m	84.1	3.18, m
5	69.9	3.78, m	72.6	3.38, m
6	18.6	1.19, d (5.6)	18.8	1.36, d (6.0)
3-OMe	58.5	3.40, s	57.6	3.39, s
	**Ole**		**Thv**	
1	102.6	4.56, br d (9.2)	104.2	4.43 (d (7.6)
2	37.5	1.40, m/2.30, m	75.2	3.23, m
3	80.3	3.35, m	86.3	3.17, m
4	83.7	3.18, m	82.9	3.30, m
5	72.5	3.35, m	72.5	3.38, m
6	18.9	1.35, d (6.0)	18.5	1.36, d (7.2)
3-OMe	57.4	3.39, s	61.2	3.60, s
	**All**		**Glc**	
1	102.2	4.70, d (8.0)	104.3	4.41, d (7.6)
2	73.5	3.29, m	75.6	3.15, m
3	83.9	3.60, m	78.3	3.23, m
4	75.0	3.16, m	71.8	3.20, m
5	71.2	3.63, m	77.9	3.31, m
6	18.6	1.21, d (6.0)	63.2	3.62, m/3.84, m
3-OMe	62.5	3.58, s		

Tig, Tigloyl; Cym, β-d-cymaropyranosyl; Ole, β-d-oleandropyranosyl; Thv, β-d-thevetopyranosyl; Glc, β-d-glucopyranosyl; All, 6-deoxy-3-*O*-methyl-β-d-allopyranosyl; *a*, axial; *e*, equatorial; *α*, atoms or groups laying below the plane of structure; *β*, atoms or groups laying above the plane of structure.

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
