# Peer review of "Five New Pregnane Glycosides from Gymnema sylvestre and Their α-Glucosidase and α-Amylase Inhibitory Activities"

_molecules, 2020, doi:10.3390/molecules25112525_

Round 1
Reviewer 1 Report
This article reports the isolation of five new pregnane glycosides from Gymnema sylvestre and their α-glucosidase and α-amylase inhibitory activities. The plane structures of the new compound given by the authors seem to be correct. However, the stereochemistry of the aglycone moiety of 1-5 and the sugar analysis of a 6-deoxy-3-O-methyl-β-D-allopyranose have to need for the more precise evidence. The authors examine α-glucosidase and α-amylase inhibitory activities of the isolated compounds. There might not be α-glucosidase inhibitory activity of the isolated compounds. In my opinion, some revision is needed to make this manuscript suitable for publication in Molecules.
Remarks:
1, How did the authors determine the stereochemistry of C-17 and C-20 of the aglycone moiety of 1-5? Please describe in the text especially about C-20.
2, In the case of acid hydrolysis of compound 2 with sugar B, it is difficult to obtain 6-deoxy-3-O-methyl-β-D-allopyranose as a monosaccharide. Disaccharide pachibiose is often obtained. Please read the paper (‘Pregnane and pregnane tetraglycoside from Marsdenia roylei’ Gupta, V. et al, Natural Product Research, 25(10), 959-973, 2011) and reconsider experiment conditions and method for the sugar analysis.
3, How did the authors identify the monosaccharides. Please describe how to get the authentic samples of the sugars (D-cymarose, D-oleandrose, D-thevetose, and 6-deoxy-3-O-methyl-β-D-allopyranose) in experimental section.
4, The authors examined the α-glucosidase and α-amylase inhibitory activities of the isolated compounds. The isolated compounds had almost no α-glucosidase inhibitory activity. I think there is no need for a description of α-glucosidase inhibitory activity.
5, Line 167, ‘to the positive control, a carbose’ should be ‘to the positive control, acarbose.’
Reviewer 2 Report
In the first part of their study, Kim and coworkers give structural proofs of five new pregnane glycosides extracted from leaves of Gymnesa sylvestre.
In the main text, precise description of the real fractions isolated that contained new compounds is required. Moreover, only two solvents seem to be used (chloroform and ethyl acetate) while three different solvent are mentioned within the experimental part (n-hexane, dichloromethane and ethyl acetate). What is true? Another point: a sophisticated series of chromatographic columns were used to obtaine the desired compounds. This should be part of the description in the main text, justifying the choice of the stationary phases.
The structural elucidation based on NMR spectra of high quality, is well written and correlation with known structures appropriated for this study, for both glycon (from tri- to tetrasaccharide) and aglycon (steroid-like) one. This corresponds to the five new compounds. However, the authors are invited to precise the compounds isolated in all other fractions given in the experimental part. What do they contain? Mixture of these new glycoconjugates or other derivatives with known structures? Moreover, since differences in the glycon part are observed and demonstrated, does the starting methanolic extract also contain pregame derivatives but with structural modulations on the lipophilic part?
The second part of the paper deals with the ability of these new glycoconjugates to inhibit alpha-glucosidase and alpha-amylase. But which ones? Saccharomyces cerevisiae and Aspergillus oryzae should clearly appear in the main text. The resulting activities were compared with the one of acarbose, a well-known antidiabetic agent.
In conclusion, many further efforts are still required to optimize the paper before final acceptation for publication in Molecules.
Minor remarks:
- Abstract: “All these new compounds were evaluated” for their…
- Tables 1 and 2: all m NMR signal should be described with margins (from a low to a larger delta-value)
- L167: acarbose in one word
- Experimental part: all ml have to be replaced by mL
Round 2
Reviewer 1 Report
This article reports the isolation of five new pregnane glycosides from Gymnema sylvestre and their α-glucosidase and α-amylase inhibitory activities. The responses to the reviewer comments are adequate, and the paper is well revised. Please check and response to the comments as below.
Comments:
The authors responded as
“Our experiment used strong acid condition [1.0 N HCl (dioxane−H2O, 1:1; divided to two phase dioxane phase and H2O phase)]. When the bond between saccharide and aglycone is broken, the saccharide will go to water phase. In water phase (contain strong acid condition), oligosaccharide is easy to form monosaccharide. “ and “In addition, acid hydrolysis method is used to confirm D or L-monosaccharide by comparing the specific optical rotation. “
According to responses for comments 2 and 3 from reviewer 1, the authors could detect all the monosaccharides (D-cymarose, D-oleandrose, D-thevetose, and 6-deoxy-3- O-methyl-β-D-allopyranose) as a result of acid hydrolysis. Please show the yields of these monosaccharides after acid hydrolysis of 1-5 in experimental section.
If possible, please show 1H and 13C NMR spectral data of these monosaccharides for Supprimentary material section. Especially, the data for 6-deoxy-3- O-methyl-β-D-allopyranose is valuable.
Reviewer 2 Report
Two minor remarks have still to be managed:
Tables 1 and 2: all m NMR signal should be described with margins (from a low to a larger delta-value)
Experimental part: all ml have to be replaced by mL
Author Response
Please see the attachment

This manuscript is a resubmission of an earlier submission. The following is a list of the peer review reports and author responses from that submission.
Round 1
Reviewer 1 Report
The ms by Van Kiem and coworkers deals with the chromatographic isolation, followed by NMR identification and characterization, of 5 pregnane glycosides. This is a specific group of molecules synthesized by a variety of Gymnema sylvestre, a plant belonging to the Asclepiadacae family having putative medicinal properties according to the Ayurvedic and folk medicine. Authors also tested in vitro the capacity of these molecules to inhibit α-Glucosidase and α-Amylase Activities. According to their inhibitory capacity on alpha-amylase, authors conclude that two, out of five molecules can be considered an expedient strategy to control glycemic response in type 2 diabetes.
The chemical identification and characterization the five glycosides has been performed according to appropriate methodologies and contributes to better knowledge of the wide spectrum of molecules synthesized by this family of plants.
On the other hand, the part concerning the inhibitory effects on alpha-glucosidase and alpha-amylase has very limited, if any, biological interest. This kind of molecules undergo a profound metabolic transformation once ingested. Therefore, the possibility that they can interact with intracellular enzymes and modulate their activity is close to zero. There is indeed a limited possibility to affect saliva alpha-amylase but the overall effect of this (putative) inhibition on diabetic dysfunctional metabolism, in real condition, is likely to be very limited, if any. This weakness is reinforced by the concentration at which these compounds exert an inhibitory capacity: these compounds would never reach 0.2 mM (!!) concentration within any biological compartment, not even in the mouth, after a reasonable quantity ingested.
This study, in my opinion, should be completely mended of the part concerning enzyme inhibition and to any concern to diabetes, the chemical characterization remaining somehow valid and novel. In this case both the introduction and the conclusions should be completely re-written.
I strongly suggest author to confirm that the 5 pregnane glycosides identified in G. sylvestre are exclusively synthesized by this specie and not common in other plants either of the same family or belonging to other families. In other words, authors should confirm the specie-specificity of the identified molecules.
Reviewer 2 Report
The manuscript describes the identification of novel compounds (glycosides) from Gymnema sylvestre and their activities as α-amylase and α-glycosidase inhibitors.
I have found several minor syntaxes mistakes. For example
- line 36, the introduction of "inhibitors" after the name of the enzymes will clarify the sentence;
- line 39: after "in addition..." include to before "reducing blood sugar...", that will be better as well.
As for the Materials and Methods, the experiments were well performed.
In Results,
- figure 3: the text says that compound 5 has the weak α-glycosidase inhibitor activity; however, from the five compounds tested compound 5 has the higher one. I understand what you want to point out, that its activity is lower than that of the control, but in order to make it clear, I suggest eliminating the article "the" before weak or indicate that it was lower than that of the control.
The conclusions are in accordance with the obtained results.
Author Response
1) The manuscript describes the identification of novel compounds (glycosides) from Gymnema sylvestre and their activities as α-amylase and α-glycosidase inhibitors.
I have found several minor syntaxes mistakes. For example
line 36, the introduction of "inhibitors" after the name of the enzymes will clarify the sentence;
line 39: after "in addition..." include to before "reducing blood sugar...", that will be better as well.
As for the Materials and Methods, the experiments were well performed.
Answer: These errors were checked and corrected as the reviewer’s comment.
2) In Results,
figure 3: the text says that compound 5 has the weak α-glycosidase inhibitor activity; however, from the five compounds tested compound 5 has the higher one. I understand what you want to point out, that its activity is lower than that of the control, but in order to make it clear, I suggest eliminating the article "the" before weak or indicate that it was lower than that of the control.
Answer: As the reviewer’s comments. This sentence was rewriten as “. Compound 5 showed the moderate α-glucosidase inhibitory activity (37.8 ± 1.5%) at the concentration of 200 μM, compared to the positive control, a carbose (inhibition percentage of 57.8 ± 3.2% at the concentration of 155 μM) (Figure 3).”
Reviewer 3 Report
The paper entitled “Pregnane Glycosides from Gymnema sylvestre and Their α-Glucosidase and α-Amylase Inhibitory 3 Activities” deals with isolation of known and new compounds from the title plant , as well as the study of their α-glucosidase and α-amylase inhibitory activities.
The structural characterization seems accurate, however, I would like to raise the following points that may contribute to improve the interest of this work:
The Results section can be shortened. Isolates are close derivative of other compounds, and thus I suggest that the authors describe some detail of the structural determination (particularly rows 77-95 of page 4).
- I suggest to add the fragmentation patterns of glycosides from MS analysis.
- α-glucosidase and α-amylase: Please replace µg/ml by µM for acarbose, which is more significant for pure compounds and commonly used.
- The compounds Retention time of isolates need to be added
In all figure legends the significance should be mentioned
The authors have to add data about silica gel amount. The authors do not mention how the fractions were collected. Were they collected by time or by volume?
Please, include this information.
Author Response
The paper entitled “Pregnane Glycosides from Gymnema sylvestre and Their α-Glucosidase and α-Amylase Inhibitory Activities” deals with isolation of known and new compounds from the title plant, as well as the study of their α-glucosidase and α-amylase inhibitory activities.
The structural characterization seems accurate, however, I would like to raise the following points that may contribute to improve the interest of this work:
1) The Results section can be shortened. Isolates are close derivative of other compounds, and thus I suggest that the authors describe some detail of the structural determination (particularly rows 77-95 of page 4).
Answer: The results and Discussion was revised as the reviewer’s comments.
2) I suggest to add the fragmentation patterns of glycosides from MS analysis.
Answer: The authors would like to thank you very much for your valuable comments. To elucidate the structure of new pregnane glycosides, we have used the chemical and spectroscopic methods. In addition, the structures of compounds were also compared to similar compounds from Gymnema sylvestre. As above idea, the aglycone was supposed to have the same configurations as those of known compounds, gymnepregoside F and 12-O-(E)-cinnamoylgymnepregoside F from G. sylvestre [12]. In addition, the alkaline hydrolysis of 1 gave sarcostin, ((20S)-3β,8β,12β,14β,17β,20- hexahydroxypregn-5-ene) [13]. The sugar moieties were proved by acid hydrolysis; sugar linkage was confirmed by HMBC correlations. Based on the above evidence, please allow used do not use MS/MS analysis of new compounds 1-5.
Please see reference of Gymnema sylvestre: Xu, R.; Yang, Y.; Zhang, Y.; Ren, F.; Xu, J.; Yu, N.; Zhao, Y. New pregnane glycosides from Gymnema sylvestre. Molecules 2015, 20, 3050-3066.
3) α-glucosidase and α-amylase: Please replace µg/ml by µM for acarbose, which is more significant for pure compounds and commonly used.
Answer: Acarbose used as a positive control. The compounds and acarbose were evaluated biological activities at the same time (acarbose has MW of 645). Please allow us convert the concentration unit of μg/ml (100 μg/ml) to the concentration unit of μM (155 μM -100/645*1000)
4) The compounds Retention time of isolates need to be added.
Answer: The retention time of compounds were added in Extraction and isolation section.
5) In all figure legends the significance should be mentioned
Answer: This was revised as the reviewer’s comment.
6) The authors have to add data about silica gel amount. The authors do not mention how the fractions were collected. Were they collected by time or by volume?
Answer: a) The silica gel amount was added in Extraction and Isolation.
- b) With open column and gradient solvent, each volume of solvent was described in Extraction and Isolation.
- c) With open column and isocratic solvent: after solvent eluent in to testube (12.5 cm length × 1.6 cm ID). Then each every five testubes wac chosen to check TLC. For examples
Figure 1. TLC from GS2F1 using solvent eluent MeOH:H2O (4:1, v/v).
Based on the above TLC (Figure 1), testube numbers from 35 to 43 were collect to give fraction GS2F1A; 44-50 (GS2F1B); 60-75 (GS2F1C; 85-95 (GS2F1D), 100-110 (GS2F1E)
Overall, the fraction GS2F1 (2.8 g) was chromatographed on a RP-18 column (60.0 cm length × 5.0 cm ID) using MeOH:H2O (4:1, v/v) as solvent to give five fractions, GS2F1A-GS2F1E.
Thus, please allow us concisely decribe with open column and isocratic solvent condition.
Round 2
Reviewer 1 Report
Authors simply ignored the point raised about the possibility that the molecules identified may have a significant role in enzimatic inhibition in vivo.
In my opinion, the study still suffers of a complete lack of scientific robustness in the part dealing with the biological effect of pregnane glycosides as a possible strategy to counter hyperglycemia.
I confirm my suggestion to completely remove the part of ms describin enzyme inhibition. This set of information has no biological relevance, is misleading and can generate false expectations.